# Contrasting social and non-social sources of predictability in human mobility

Zexun Chen [1,2], Sean Kelty[3], Alexandre G. Evsukoff [4], Brooke Foucault Welles[5], James Bagrow [6], Ronaldo Menezes [1✉] & Gourab Ghoshal [3,7✉]

Social structures influence human behavior, including their movement patterns. Indeed, latent information about an individual's movement can be present in the mobility patterns of both acquaintances and strangers. We develop a "colocation" network to distinguish the mobility patterns of an ego's social ties from those not socially connected to the ego but who arrive at a location at a similar time as the ego. Using entropic measures, we analyze and bound the predictive information of an individual's mobility pattern and its flow to both types of ties. While the former generically provide more information, replacing up to 94% of an ego's predictability, significant information is also present in the aggregation of unknown colocators, that contain up to 85% of an ego's predictive information. Such information flow raises privacy concerns: individuals sharing data via mobile applications may be providing actionable information on themselves as well as others whose data are absent.

[1] BioComplex Laboratory, Department of Computer Science, University of Exeter, Exeter, UK. [2] University of Edinburgh Business School, Edinburgh, UK. [3] Department of Physics and Astronomy, University of Rochester, Rochester, NY, USA. [4] DCOPPE, Federal University of Rio de Janeiro, Rio de Janeiro, Brazil. [5] Northeastern University, Boston, MA, USA. [6] Department of Mathematics & Statistics and Vermont Complex Systems Center, University of Vermont, Burlington, VT, USA. [7] Department of Computer Science, University of Rochester, Rochester, NY, USA. ✉email: r.menezes@exeter.ac.uk; gghoshal@pas.rochester.edu

The recent availability of extensive geolocated datasets related to human movement, has enabled the quantitative study of human movement at an unprecedented level[1], contributing greatly to insights in estimating migratory flows, traffic forecasting, urban planning, mitigating pollution, uncovering socioeconomic inequalities, and epidemic modeling among other applications[2–11]. Several common regularities have been observed across these studies, including bursty activity rates, tendencies to visit a select few locations disproportionately more than others, as well as decreasing likelihood to explore as time goes on[12–17]. A related aspect that can enhance the potential of these findings, particularly for urban planning and the control of epidemics, is the ability to predict the future locations of individuals or groups using their prior history of travel. Indeed, it has been shown that a perfect algorithm can predict, with between 70 and 90% certainty, an individual's future location given their prior location visits[18], depending upon the spatiotemporal granularity of observations[19].

Human beings are typically highly social creatures and social structures can influence behavior in a variety of human activities including movement patterns. In fact, it has been shown that social relationships statistically account for between 10 and 30% of all human movement[20]. Social structures inherently encode information flow between parties, such that residual information about an individual can be inferred from their social ties. Such a phenomenon was demonstrated in the context of online interactions, where about 95% of an individual's potential predictive accuracy was contained in their social network, despite no recourse to information about the person in question[21]. Coupled with the observation that movement patterns in the virtual and physical domains are strikingly similar[22], this leads to the intriguing question as to whether one can leverage a person's social network to predict their future mobility patterns, absent any information on their own history. This possibility holds promise for a number of applications, and may be particularly relevant in the context of mitigating future pandemics[23,24], where a key tool in the arsenal is contact tracing based on mobility patterns[25,26]. However, accurately mapping human mobility can be challenging due to understandable privacy concerns and people's willingness to disclose or share personal data[27,28].

Location-based social networks (LBSNs) and call detail records (CDRs) from mobile phones yield opportunities to examine social relations to human mobility, containing information both about sequences of location visits and (in some cases) information on the underlying social network. At the same time, spatially aggregating these data can reveal individuals in different social circles who visit similar or overlapping locations; for instance, people working in the same building but with different companies, or parents whose children attend the same schools but are unknown to each other. These non-social ties are potential predictors of a person's mobility trajectory. Terming such individuals "non-social colocators", we ask whether and to what extent do such colocators yield predictive mobility information, and how this information compares to that of social ties?

Here we apply non-parametric information-theoretic estimators to study human mobility extracted from three LBSNs, that contain sequences of location trajectories and the (reported) social network of a subset of users. In addition, we also analyze CDRs from Rio de Janeiro in Brazil. Each type of dataset has its own limitations: in the case of LBSNs, the mobility patterns being biased by the types of individuals using the platform; in the case of CDRs, the lower-spatial resolution. Despite this, we demonstrate the existence of information transfer in all four networks, finding that a given ego's future location visits can be predicted, with between 80–100% of the ego's own accuracy, by studying the historical patterns of just 10 of their alters (ranked by number of

common locations visited). Remarkably, non-social colocators, while individually providing less information than social ties, can in the aggregate provide similar levels of predictability. The information flow provided by colocators is also surprisingly robust to temporal-displaced colocations, implying users that never physically colocate can still provide comparable information to social ties. Indeed, the information transfer appears to be driven by the overlap of unique locations visited by the ego and alters, in both social ties and non-social colocators.

## Results

**Mobility data.** Our study uses three publicly available datasets and one private call record dataset that contain mobility traces, with the former containing social networks of a subset of the users of the platforms. The first is BrightKite, a location-based social networking service (LBSN)[20,29] containing 4,491,143 geotagged check-ins by 58,228 users over a period of April 2008–October 2010. The second dataset is from Weeplaces, a website that generated visualizations and reports from location-based check-ins in platforms such as Facebook and Foursquare[22]. The considered data contains only Foursquare check-ins, which includes 7,658,368 geotagged check-ins produced by 15,799 users from November 2003–June 2011. We also include an analysis on Gowalla[20], another LBSN consisting of 6,442,890 check-ins by 196,591 users over a period of February 2009–October 2010. Finally, we consider a CDR dataset collected in the Rio de Janeiro Metropolitan Area (RJMA), Brazil[30], consisting of 22,116,252 call records by 35,338 users pinging 1835 cell antennas over the period of January 2014–June 2014. (For more details, see "Methods" and Section S1.)

In Fig. S1 we show the check-in maps for each of the datasets indicating global coverage with the highest concentrations in North America and Western Europe. In Fig. S2 we plot the distributions for the number of distinct locations visited by users, the jump-length, and the radius-of-gyration. The LBSNs show similar trends for their jump-length and radius-of-gyration, consistent with other sources of mobility data[1]. Differences exist, however, in the number of unique locations visited. BrightKite, in particular, contains a large fraction of users that visit only a few distinct locations (between 1 and 3). The CDR data differs from the LBSN in having much sharper cut-offs in the tail of the distribution, a consequence of scale—trips are bounded by the spatial area of the city, unlike in the LBSNs that contain inter-city and international travel. The statistical trends, however, are consistent with that seen in other CDR datasets[31].

**Information contained in egos.** We begin our analysis by examining the information contained in the location trajectories of all egos in each of the datasets; this serves as a baseline when comparing information flow with social ties and non-social colocators. The degree of uncertainty in capturing the future locations of a trajectory $A$, given past observations, is encoded in the entropy rate $S_A$ of the trajectory. Accounting for both frequency of location visits, as well as temporal ordering (specific ordered sequences in the data), we make use of a non-parametric estimator[32,33] given by the expression

$$\hat{S}_A = \frac{N \log_2 N}{\sum_{i=1}^{N} \Lambda_i},\qquad(1)$$

where for a trajectory $A$ of $N$ moves of an individual, $\Lambda_i$ is the length of the shortest trajectory sub-sequence beginning at position $i$ not seen previously, and the entropy is measured in bits. This estimator has been applied to mobility patterns and online social activities[18,21]. In the absence of any temporal structure in the sequence, the expression converges to the standard

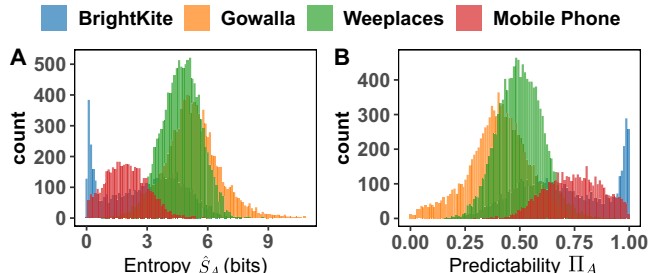

**Fig. 1 Entropy and Predictability in the four mobility datasets. A** The distribution of the entropy $\hat{S}_A$ (Eq. 1) for each of the four datasets. **B** The corresponding distribution of predictability $\Pi_A$, calculated by inverting Eq. 2, tells us how well an ideal algorithm can predict an individual's future location given their mobility history.

Shannon entropy[21]. In Fig. 1A, we plot $\hat{S}_A$ for the datasets, finding peaks between 4 and 5 bits with varying degrees of spread for Gowalla and Weeplaces. For BrightKite we find a peak at ≈ 1 bit, a reflection of the fact that a large fraction of users visit only between 1 and 3 distinct locations and therefore have low entropy. The CDR data is peaked at a lower value (≈2) bits than either Gowalla or Weeplaces, but higher than BrightKite. This is primarily for two reasons: the possible space of locations that can be visited is bounded by the city-size and is much lower than in the other datasets (which include long-range travel). Second, the spatial resolution is much lower than in the LBSN. A location corresponds to the coverage area of a cell-phone antenna, which varies from about a city-block to a neighborhood, and therefore locations that might be distinct in Weeplaces, for instance (coffee shops, restaurants, banks, etc.) are coarse-grained into a single location[18,31].

An intuitive measure to interpret these results is the perplexity $2^S$: we are as uncertain about future visits for a trajectory with entropy rate 3 bits, for example, as we would be when choosing uniformly at random from $2^3 = 8$ possible locations. Using $\hat{S}_A$ from Fig. 1A, this implies that on average knowing the past history of the typical ego allows us to reduce the possible number of future location visits to between 16 and 32 sites. Given the average number of distinct locations (Fig. S2) that a typical user visits across the datasets (107 total distinct locations per user on average in BrightKite, 198 in Gowalla, 213 in Weeplaces, and 38 in the CDR), information due to the spatiotemporal regularities of ego trajectories represent an order of magnitude reduction from choosing across all locations uniformly at random, a result consistent with that found in other mobility studies[18,19].

The entropy rate can also be interpreted using Fano's inequality[34] to define the predictability $\Pi_A$, the upper bound of how often an ideal predictive algorithm can correctly guess the next location visit, given prior history. This predictability is calculated by inverting

$$\hat{S}_A \leq H_A(\Pi_A) + (1 - \Pi_A)\log_2(n-1), \qquad (2)$$

where $n$ is the number of distinct locations visited and $H(x)$ is the binary entropy function capturing the entropy of a simple Bernoulli trial (in this case achieving maximal predictability or not). Utilizing $\Pi_A$ allows us to leverage information theory to mathematically bound the performance of all real predictive methods given an information source inferred uncertainty. Figure 1B shows the distributions of predictability, finding differences across the four datasets that reflect their respective distributions of $\hat{S}_A$. While BrightKite shows a distinct spike of highly predictable ($\Pi_A \approx 1$) users (as a consequence of the low-entropy users), in Gowalla, $\Pi_A$ is peaked at ≈ 40%, with a wide

spread around the peak. This is due to the fact that some users visit many locations (indeed, 23 users never return to a previously visited location), resulting in a high entropy rate, and low predictability, a feature that likely stems from Gowalla incentivizing its users to discover new locations (see Section S1). In Weeplaces, $\Pi_A$ is peaked at ≈ 50% with a tighter bound around the peak as compared to the other datasets. Finally, in the CDR data, we find a peak at ≈ 75% with a spread comparable to that seen in Weeplaces and within the range found in other CDR datasets[31]. The results suggest that each dataset has its own peculiarities. BrightKite contains a large fraction of users that do not visit many distinct locations; in contrast, Gowalla has users that are incentivized by the nature of the platform to sample as wide a location-set as possible. The CDR data has low spatial resolution, and of course all datasets are biased by the behavior of the population they cover. This observed diversity in mobility behavior across the four platforms, thus provides a stringent robustness check on the results to follow.

**Information contained in social ties and non-social colocators.** Next we examine information flow, how much mobility information about the ego is contained in the sequence of location visits of their alter(s), absent any information about the ego's own location history. We do so by analyzing the social and non-social colocation networks in each of the datasets (see "Methods" for details of the construction). The information flow is measured by the cross-entropy[21,35], which is greater than the entropy when the alter contains less information on the ego than the ego itself, and quantifies information loss when we have access to only the alter's past. To estimate the cross-entropy between two sequences, Eq. 1 can be modified to account for $A$ and $B$ (representing the mobilities of the ego and alter, respectively) according to

$$\hat{S}_{A|B} = \frac{N_A \log_2(N_B)}{\sum_{i=1}^{N_A} \Lambda_i(A|B)}, \qquad (3)$$

where $N_A$ and $N_B$ are the lengths of the sequences $A$, $B$, and the cross-parsed match length $\Lambda_i(A|B)$, is the length of the shortest location sub-sequence starting at position $i$ of sequence $A$ not previously seen in sequence $B$. Here, 'previously' refers to those locations $\ell_j$ in sequence $B$ with $t_j < t_i$, the timestamp of the check-in location $\ell_i$ in sequence $A$.

As with the cross-entropy, one can generalize the predictability $\Pi_A$ to the cross-predictability $\Pi_{A|B}$ by applying Eq. 2 to Eq. 3. For the remainder of this paper, both social ties and non-social colocators have been processed by retaining alters that provide better-than-random information about their ego, as well as removing from the colocation network any spurious colocators (see Sections S2.1, S7).

Figure 2 shows the results of our information metrics on the Weeplaces dataset. Panels A and B show the distribution of the cross-entropy and predictability for the rank-1 social tie and non-social colocated alter. We see that the top social tie provides slightly more information than the top colocator, with predictability slightly right-skewed (Fig. 2B). While social ties provide more predictive information, the distribution also shows the existence of some non-social colocators that provide mobility information comparable to that provided by social ties. Furthermore, the predictability of egos are positively correlated with the predictability of their top alter (Fig. S9), meaning highly predictable egos tend to have highly predictable top alters, and similarly more unpredictable egos tend to have less predictable alters.

We have thus far looked at the individual and pair-wise information in ego-alter pairs, a limited analysis, given that these are being considered as information sources in isolation. Next,

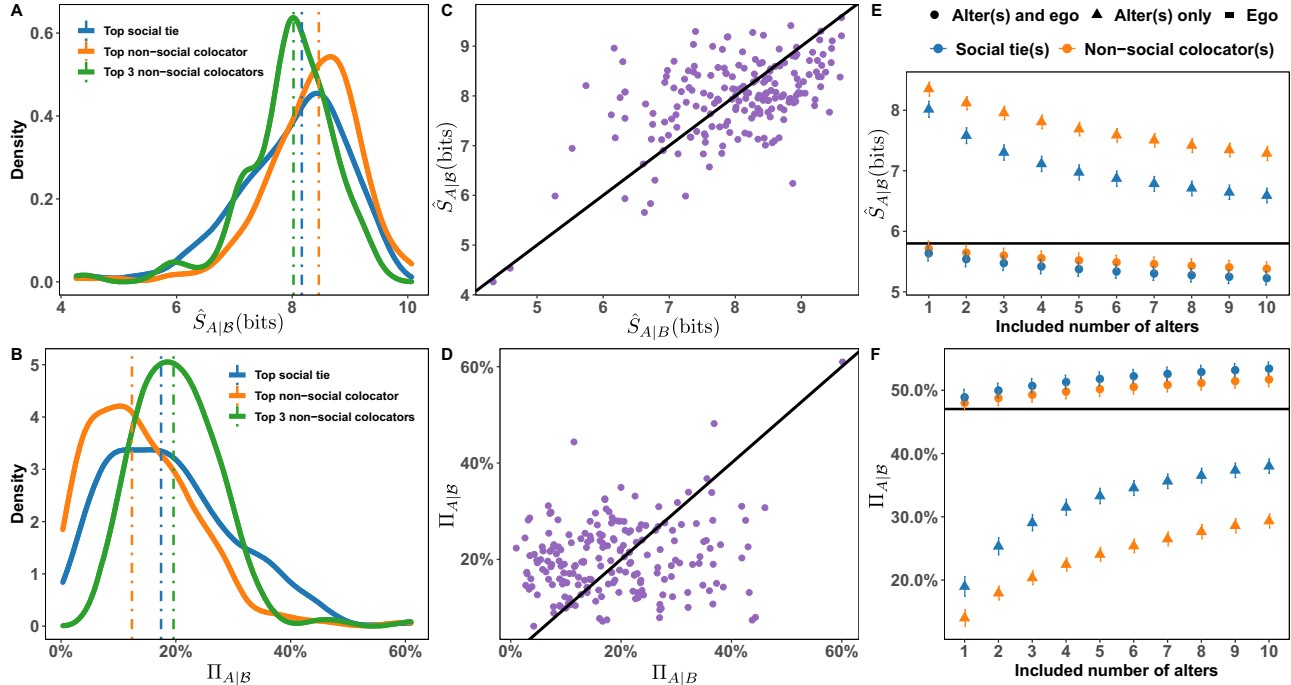

**Fig. 2 Cross-entropy and predictability in social ties and non-social colocators. A** Distributions of $\hat{S}_{A|B}$ for the rank-1 social tie (median 8.17 bits), non-social colocator (median 8.46 bits), and $\hat{S}_{A|\mathcal{B}}$ for the top-3 non-social colocators (median 8.02 bits) in Weeplaces. **B** The corresponding $\Pi_{A|B}$ for the social (median 17.43%), and non-social colocators (median 12.35%), and $\Pi_{A|\mathcal{B}}$ for the top-3 non-social colocators (median 19.60%). **C** $\hat{S}_{A|B}$ encoded in the top-social tie as a function of $\hat{S}_{A|\mathcal{B}}$ for the top-3 non-social colocators. Each point corresponds to a single ego and the solid line denotes $y = x$. **D** As in panel (**C**) but with predictability instead of cross-entropy. **E**, **F** $\hat{S}_{A|\mathcal{B}}$ and $\Pi_{A|\mathcal{B}}$ after accumulating the top-ten social alters and non-social colocators. Horizontal lines denote the average entropy (5.80 bits) of egos and their self-predictability (47.05%). Shapes indicate whether the past trajectory of the ego was included in the sequence (circles) or excluded (triangles). Error bars denote 95% CI.

we examine the information content of a multiplicity of an ego's alters, or in other words examine the information content of a subset of the ego's colocators by adapting the cross-entropy to a set of alters. To estimate the amount of information needed to encode the next location of sequence $A$ given the location information in a set of sequences $\mathcal{B}$, we generalize the pairwise cross-entropy to the cumulative cross-entropy according to[21], thus,

$$\hat{S}_{A|\mathcal{B}} = \frac{N_A \log_2(N_{A\mathcal{B}})}{\sum_{i=1}^{N_A} \Lambda_i(A|\mathcal{B})}, \qquad (4)$$

where $\Lambda_i(A|\mathcal{B}) = \max\{\Lambda_i(A|B), B \in \mathcal{B}\}$ is the longest cross-parsed match length over any of the sequences in the set of sequences $\mathcal{B}$, $N_{A\mathcal{B}} = \sum_{B \in \mathcal{B}} w_B N_B / \sum_{B \in \mathcal{B}} w_B$ is the average of the lengths $N_B$, and $w_b$ is the number of times that matches from sequence $A$ are found in each sequence $B \in \mathcal{B}$. Note that if there is only one sequence in $\mathcal{B}$, Eq. 4 reduces to Eq. 3. By applying Fano's inequality (Eq. 2), we denote the corresponding cumulative cross-predictability as $\Pi_{A|\mathcal{B}}$. In Fig. S5 we see that on average, alters associated with more frequent colocations contain more information content: as a consequence, we rank alters according to frequency of colocations. Plotting the average number of colocations between ego-alter pairs saturate at around 10 alters in all four datasets (Fig. S4), and therefore we examine the information content of the top-10 most frequently colocated social alters and non-social colocated alters. For a fair comparison between these two different sources of mobility information, we focus on egos in both the social and colocation network with at least ten alters in each network, leading to 33 (BrightKite), 97 (Gowalla), 199 (Weeplaces), 484 (Mobile Phone) egos (cf. third column of Table S2 for details).

By moving from one non-social colocator to three, we see in Fig. 2A, B that considerably more predictive information is present, with the peak of $\Pi_{A|\mathcal{B}}$ shifted significantly rightward. Further, the peak is now at a higher value than the peak for the top social tie (Fig. 2B), indicating that many egos are better predicted by three non-social colocators than they are by their top social tie. We further emphasize this relationship in Fig. 2C, D with scatter plots comparing the cross-entropy of the top social tie to the cumulative cross-entropy of the top-3 non-social colocators; any points above the line $y = x$ in panel D demonstrate more information flow from the colocators about the ego than from the top social tie. Individually, non-social colocators are less informative than social ties, but collectively they can meet or exceed the information content of individual social ties.

Expanding on the comparison between social ties and non-social colocators, in Fig. 2E, F, we plot the cumulative cross-entropy and cross-predictability $\Pi_{A|\mathcal{B}}$, finding a progressive increase in predictability as we accumulate more alters (positive Spearman's $\rho$ across 88.94% all users, $p < 0.05$). A given number of social ties provides more information on average than the same number of non-social colocators, as demonstrated by the lower curve in entropy in panel E and higher curve in predictability in panel F. Specifically, 94.47% of egos in Weeplaces show significantly higher social tie predictability than non-social colocator predictability (paired one-sided $t$-test, $p < 0.01$). However, while the colocator curve in Fig. 2F sits below the social curve for a given number of alters, we do see that on average a greater number of colocators can exceed the information content of a small number of social ties. For instance, the top-3 non-social colocators provide higher predictability than the top social tie, and the top-7 colocators provide higher predictability than the

top-2 social ties. The corresponding results for the other three datasets are shown in Fig. S6 (BrightKite), Fig. S7 (Gowalla) and Fig. S8 (CDR). While the values of the entropies, predictability, cross-predictability and the comparable information contained in the top social tie viz the number of non-social colocators vary between the datasets (reflecting their inherent biases), in all cases the trends are consistent with that seen in Fig. 2.

**Combined information of both ego and alter's past.** While alter information about the ego is important to understand, especially for matters of privacy (see "Discussion"), we also wish to understand whether that information is redundant when given the ego's past. We, therefore, show in Fig. 2E, F curves including the ego's past alongside that of the alters. We find that non-redundant information exists in both types of alters, with a gain of $\approx 10\%$ predictability for the top-10 social ties and $\approx 14\%$ for the top-10 non-social colocators. Fig. 2F also demonstrates that both curves appears to saturate as more alters are included, an effect observed in other studies[21]. This saturation effect is examined in Section S4, where $\Pi_{A|B}$ is extrapolated beyond our data window of 10 alters by fitting a nonlinear saturating function and estimating the extrapolation predictability $\Pi_\infty$. For Weeplaces we find $\Pi_\infty = 44.32$ and 39.79% for social ties and colocators, respectively. Compared to the average $\Pi_{ego} = 47.05\%$ of all egos in the network, this means that 94 and 85%, respectively, of the potential predictability of an ego is in principle available in that ego's alters. Including the ego's past trajectory, for Weeplaces we find $\Pi_\infty = 56.70$ and 56.25% for social ties and colocators, respectively. Compared to the average $\Pi_{ego} = 47.05\%$ of all egos in the network, this means an additional 19.5–20.5%, respectively, of the potential predictability of an ego is in principle available when including its alters. The closeness of these values underscore the high degree of predictive information available in the non-social colocators. The corresponding findings for the other three datasets are shown in Table S3. In all cases, we see a gain of predictability in the ego when alters are included in the range of 2–10%.

Extrapolation analysis demonstrates the overall relative value of non-social colocators, but it does not allow us to determine more precisely how many non-social colocators equal the information content for a given number of social ties. Therefore, to better quantify the relative information content provided by social ties compared with non-social colocators, we examine the predictability ratio $\Pi_{ego|social\ tie(s)}/\Pi_{ego|non-social\ colocator(s)}$ across all datasets. In Fig. 3A we present the distributions of predictability ratio comparing the top non-social colocator to the top-$k$ social ties ($k = 1, 2, 3$). For the top social ties ($k = 1$) we see that BrightKite colocators provide the closest information with a ratio just below 2, meaning the social tie provides approximately twice the predictability of the colocator. In Gowalla and Weeplaces, the difference is even stronger, with the top social ties providing approximately three times the predictability of the top colocator. Moving from the top colocator to multiple colocators, in Fig. 3B we plot the predictability ratio for increasing numbers of non-social colocators; when this curve crosses the horizontal line at a ratio of 1, we have equal amounts of information. For example, examining the first panel in B, we see that for BrightKite this happens between 1 and 2 colocators, between 7 and 8 for Gowalla, between 3 and 4 for Weeplaces and between 1 and 2 for the CDR data. This suggests that three Weeplaces colocators are equivalent to the top social tie, while in Gowalla that number is between 7 and 8. In BrightKite, a dataset characterized by high predictability and low entropy, one added social tie provides the same degree of information as two non-social colocators. Across all datasets, we see that an aggregate of fewer than 10 non-social colocators can equal the information of

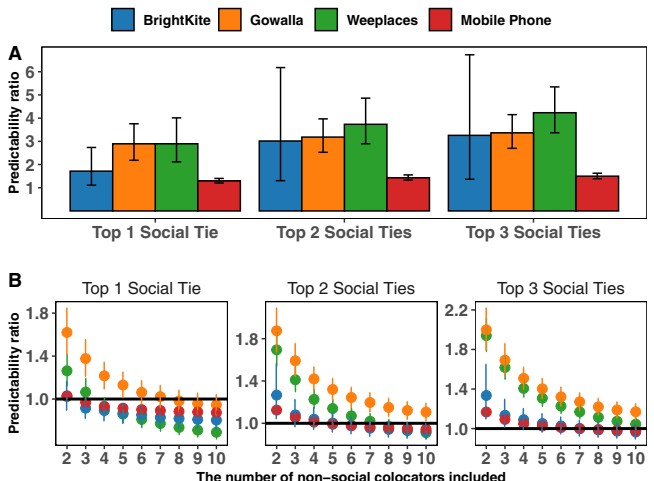

**Fig. 3 Quantifying the predictive information aggregated from non-social colocator(s) with respect to social ties. A** The predictability ratio $\Pi_{ego|social\ tie(s)}$/$\Pi_{ego|non-social\ colocator(s)}$ between the top non-social colocator and (left-to-right) the top, top-2, and top-3 social tie(s). **B** The predictability ratios between the top 2 and 10 non-social colocators and the top, top-2, and top-3 social tie(s). Error bars denote 95% CI.

the top social tie. While more colocators are needed to equal the aggregate of the top-2 or top-3 social ties, the observed decreasing trends suggest a convergence in the amount of information contained in either flavor of tie.

**Underlying spatial and temporal mobility characteristics.** We next determine the key factors that determine the near identical types of information transfer in both types of ties, despite having no overlap in the pair-wise connections. One of the possibilities driving the quantity of information on the ego provided by alters is the information inherent in the locations themselves. That is, it is reasonable to surmise that information about the ego is derived from shared visits to common locations, given that predictability of the ego itself depends on the patterns of location visits in their trajectory. While alters do not necessarily visit all the locations that their egos do, nor would they necessarily visit at the same time, one can hypothesize that higher-ranked alters share more distinct locations with the ego than lower-ranked ones. If the trend is similar across both social ties and non-social colocators, then this would be a plausible mechanism for the similarity in the observed cross-predictability.

To measure this, we compute the proportion of unique locations visited by the ego and its alters. For an ego $A$ we define the Overlapped Distinct Location Ratio (ODLR) $\eta$ as the fraction of $A$'s visited locations also visited by an alter $B$. Formally,

$$\eta_{A|B} = \frac{|Y_A \cap Y_B|}{|Y_A|} \quad (5)$$

where $Y_A$ and $Y_B$ are the sets of locations visited by $A$ and $B$, respectively, and $|\cdot|$ denotes set cardinality.

In Fig. 4A we plot the ODLR as a function of alter rank. As alters are ranked according to the frequency of overlap of any location visit of the ego, as opposed to distinct location visits, there is no reason to a priori expect that a rank-1 alter will share the most number of distinct locations in their trajectory with the ego. Nevertheless, that is indeed what is observed across all datasets, with a monotonically decreasing trend of ODLR as one considers lower-ranked alters. This monotonic trend is considerably stronger for social ties than for non-social colocators,

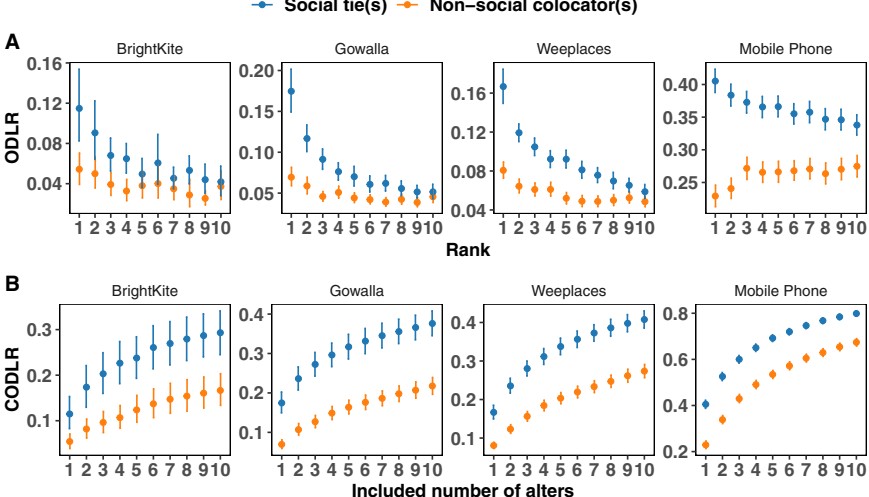

**Fig. 4 The degree of ego–alter distinct location overlap. A** The Overlapped Distinct Location Ratio (Eq. 5) indicates that higher ranked alters share more unique location visits than lower-ranked ones, with the top (rank-1) alter showing the most shared location. The trend is stronger for social ties than non-social colocators. **B** The Cumulative Overlapped Distinct Location Ratio (Eq. 6) shows increasing discovery of unique locations in the ego's trajectory as alters are added in order of decreasing rank, but that the rate of discovery slows. Error bars denote 95% CI.

although the difference diminishes with the number of alters added.

The ODLR fails to consider locations shared across multiple alters, instead focusing on one alter at a time. Yet we previously saw (Fig. 2) the importance of examining the aggregate information, particularly when comparing non-social colocators to social ties. Therefore, we generalize the ODLR to a Cumulative Overlapped Distinct Location Ratio (CODLR) by taking the union of the location sets of multiple alters according to,

$$\eta_{A|\mathcal{B}} = \frac{|\cup_{B\in\mathcal{B}}(Y_A \cap Y_B)|}{|Y_A|}, \qquad (6)$$

where $A$ is the ego and $\mathcal{B}$ is the set of all alters. We plot the results in Fig. 4B finding similar increasing monotonic trends across datasets and networks. As alters are added, more information on the ego's unique locations are discovered, saturating at between 30 and 40% after 10 social ties, and between 15 and 30% for non-social colocators. The low spatial resolution of the CDR data makes it an outlier, in that the saturation occurs at 80 and 65% respectively. Nevertheless, across all datasets, we observe that larger numbers of colocators provide comparable location overlap as a smaller number of social ties, emphasizing both the relative importance of social ties and the extent of useful information present in the aggregation of non-social colocators.

We connect ODLR and information flow directly in Fig. 5, by plotting $\eta_{A|\mathcal{B}}$ against $\Pi_{A|\mathcal{B}}$ for the top-10 alters in both types of networks, observing a strong, approximately linear trend (Pearson's $R \approx 0.66$ for social ties; $R \approx 0.67$ for non-social colocators; both significant, $p < 0.001$). Disentangling the plots by progressively adding alters from rank-1 to rank-10 shows a monotonically increasing trend for the correlations Figs. S10 and S11. The corresponding results for BrightKite (Figs. S12 and S13), Gowalla (Figs. S14 and S15), and CDR (Figs. S16 and S17) reveal similar trends.

The observed connection between information transfer and location overlap behooves one to ask whether temporal effects are a key factor. In other words, our choice of colocation is based on the simple idea that individuals in the same place at the same time contain information about mobility patterns of each other. We can relax this condition and also consider individuals that visit the

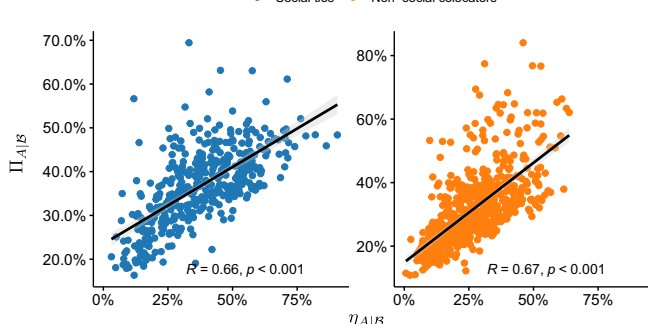

**Fig. 5 Connecting location overlap to information transfer.** Regression analysis of cumulative cross-predictability $\Pi_{A|\mathcal{B}}$ and CODLR $\eta_{A|\mathcal{B}}$ for the Weeplaces dataset with the top-10 alters. Here $R$ is Pearson's correlation coefficient. The solid lines are linear regressions.

same locations as the ego, but displaced in time. For example, residents of a neighborhood can stop at their local corner store at different times of day and never run into each other, but their visits are always five hours apart because of their respective work schedules. We can investigate this by creating networks of time-displaced colocators, where now an ego and alter colocate if an alter visits a location in the time windows $[T, T - 1/2]$ hours prior or $[T - 1/2, T]$ hours following an ego visitation at the same location (see the illustration in Fig. 6A). Note that $T = 1/2$ h yields the fully connected window $[-1/2, 1/2]$.

Each network resulting from the different temporal lags will generally have different ego-alter pairs, and the set of egos with at least 10 alters may change. We consider then the common egos who have at least 10 alters for all networks constructed with temporal lags in the range $[0.5h - 12h]$ with 30 min intervals. Given this condition, we are left with Weeplaces and the mobile phone data, given that no common egos in either BrightKite or Gowalla have at least 10 alters in all temporal intervals. Nevertheless, Weeplaces and the mobile phone data are sufficiently different for checking the robustness of such an effect. We examine the trend in $\Pi_{A|\mathcal{B}}$ as a function of the temporal-lag $T$ in Fig. 6B (Weeplaces dataset) and Fig. S18 (Mobile Phone).

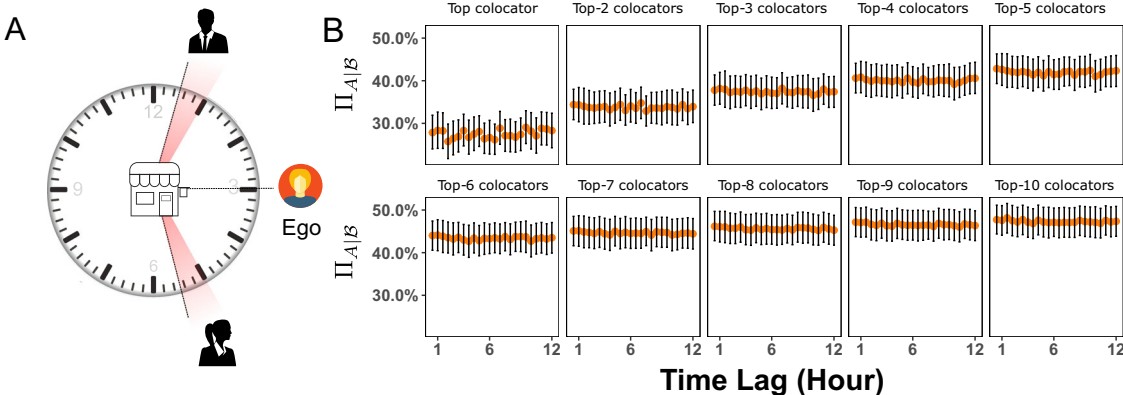

**Fig. 6 Temporal stability of non-social colocator(s) information. A** An example of two time-displaced colocators who visit the same location as the ego on a $T = 2.5$ h time lag. (Vector Clip-art designed by https://www.freepik.com.) **B** The (cumulative) cross-predictability influence of temporal-lag for non-social colocator(s) in Weeplaces. Each point corresponds to a colocation network resulting from the amount of temporal offset between an ego and alters visit to a common location. Error bars denote 95% CI.

As before, in both cases, there is an increase in both the cross-predictability $\Pi_{A|B}$ and the ODLR $\eta_{A|B}$ (Fig. S20C) as alters are progressively added, yet for a given number of alters, there is little-to-no difference between any of the networks in terms of their information content within the investigated temporal ranges. This can be seen clearly in Figs. S19 and S20A for both datasets where there is no measurable difference in predictability as a function of the various temporal-lag intervals, apart from a slight decrease in predictability as one increases the offset (colocators displaced by an hour contain marginally more predictive information to those displaced by a day). Remarkably this is despite the fact that the ego-alter overlap between these networks in Weeplaces (Fig. S19B) and the mobile phone network (Fig. S20B) indicates that colocated alters are not necessarily the same as time-displaced colocated alters. This suggests potentially unexpected sources of mobility information, as non-social colocators in both datasets do not necessarily have to be visitors of the same location at the same time to provide predictive mobility information about the ego.

## Discussion

Using information-theoretic measures, we analyzed the spatio-temporal structure of the mobility trajectories of a set of users in three publicly available LBSNs and CDR data from one private record of mobile phone users. Entropy measures were used to quantify the sequential information contained in a user's physical trajectory which revealed differences in our datasets based on the context of how users used the apps, the extent of coverage, and the differences in spatial resolution. Using these measures, we then compared the information present in the mobility patterns of an individual's (the ego's) social ties compared with non-social colocators, other users who frequently visited the same locations as the ego. Across datasets, we found the importance of social ties: consistently more information about the ego's future location was present in the past locations of the social ties than in the past locations of the non-social ties, and this held when aggregating information from multiple number of alters. Interestingly, however, this implies something important: that groups of many non-social colocators can in principle provide as much information as a smaller set of social ties, meaning that non-social sources of mobility information are in principle available. If access to social data is limited, these non-social data may, in the aggregate, be used as a replacement. A future study on the mobility information of an individual carried by non-social colocators should consider the possibility that social-economics or demographics could play a role. For instance, people working in the same building but for different companies are likely to share similar social-economic status, likewise, parents taking their children to schools may share similar household responsibilities. The extent to which these shared factors affect the predictability of a ego mobility can be investigated in richer datasets.

Each considered dataset has its own biases related to the behavior of the users, the extent of coverage, and the spatial coverage. The study relied on observational data taken, which introduces crucial caveats. In particular, the social ties reported in the datasets are incomplete reflections of a person's full social circle, and the nature of such ties may differ in the online and offline domains. Likewise, not all locations visited by an individual are recorded in these social networks, which rely primarily on user check-ins, so we expect mobility trajectories to be under-sampled as well. Followup work, including richer, more detailed data and even experimental studies, are needed to address these concerns, yet our robustness checks, including observing consistent trends across datasets with varying degrees of coverage, resolution, behavioral differences in the observed populations, and across sampling criteria, already provide rather strong evidence for the robustness of our results.

The presence of predictive information, both socially and otherwise, has crucial implications. Privacy protections regarding social data are important to protect sensitive information about a user and their social ties. Social information flow suggests that an individual's future movements can be predicted by studying the mobility patterns of a few acquaintances. On the other hand, our study also demonstrates that social ties are not the only source of predictive mobility information, and measures of colocation are enough to uncover novel sources of mobility information. This means that locations monitoring individual visits, for example, a grocery store tracking the smartphones of shoppers[36], may in principle be collecting the building blocks of mobility profiles, and individuals providing access to their mobility data may also be providing information about both social and non-social ties[37–39].

While these data can inform important applications such as contact tracing in the early stages of a disease outbreak, significant ethical concerns surrounding such information sources make it critical to place strong access constraints on mobility information. Indeed, the results presented here provide further impetus to the ongoing debate on best practices for privacy protection, both in terms of legislation and ethical algorithmic development.

## Methods

**Data structure and filtering**. In our four datasets, each event, i.e., an instance of a location visit, is timestamped and tagged with a unique location ID. In all datasets, a location visit $v$ is represented by a tuple $v = (u, \ell, t)$, meaning a user $u$ visited a location $\ell$ at time $t$. At a user level, a trajectory composed of $N_u$ discrete observations is characterized by a sequence of $N_u$ location-time pairs $(\ell_i, t_i)_{i \in 1 \dots N_u}$ where $\ell_i$ stands for the location visited at step $i$ and time $t_i$. A user $u$ who visits $N_u$ locations in total, visits $n_u \leq N_u$ distinct locations, with equality holding only if $u$ never visits a location more than once. To filter out spurious activities, we exclude inactive users and discard records with missing attributes. Furthermore, for purposes of statistical significance, we discard users who have logged $N_u < 150$ check-ins (see Section S1.3 and Fig. S3). After filtering, we are left with 510,308 events across 6132 users in BrightKite, 924,666 events across 11,533 users in Weeplaces, 850,094 events across 9937 users for Gowalla, and finally 1,382,626 events (call-records) between 4415 users for the CDR dataset. (cf. Table S1 for further details).

**Constructing social and non-social colocation networks**. Each of the LBSN datasets has social networks collected by their respective API's (details in Section S1.1), however, not all users log check-ins. Given that our goal is to examine information transfer in social networks, as it relates to location visits, we focus only on users with logged location-trajectories. To quantify the information provided by colocated non-social ties, we construct colocation networks where a tie is included between an ego and alter if they checked in at the same location within a particular time window (see Section S2 for details on egocentric network construction). We assume that individuals who colocate more often contain more predictive information about one another's whereabouts (see Section S7.4), so the ranking criteria is based on the frequencies of colocations (see Section S2.2, Figs. S4 and S5). All results presented in the main manuscript correspond to a one-hour temporal bin (results are robust to varying temporal frames, Fig. S22).

While the LBSN datasets contain explicit social network information, the social network needs to be inferred for the mobile phone data. Following the methodology outlined in[40], we consider reciprocal ego-alter pairs as social-ties, where reciprocity refers to the fact that both egos and alters exchange phone-calls. We make the reasonable (and stringent) assumption that a reciprocal call should occur at least once a week, and therefore set a minimum threshold of 30 reciprocal calls (covering the period of data-collection) in order to be considered a social-tie. For non-social colocator networks, we require that zero calls exist between the ego and alter. For sake of convenience, we use social-tie and non-social colocators to distinguish between the two types of networks. In all other respects, the construction of the networks is the same as in for the LBSNs. The summary statistics for the two types of networks in each dataset is shown in the first two columns of Table S2.

**Reporting summary**. Further information on research design is available in the Nature Research Reporting Summary linked to this article.

## Data availability

The BrightKite data used in this study are available in the SNAP database under accession code Brightkite [https://snap.stanford.edu/data/loc-brightkite.html]. The Gowalla data used in this study are available in the SNAP database under accession code Gowalla [https://snap.stanford.edu/data/loc-gowalla.html]. The Weeplaces data used in this study are available on the website [https://www.yongliu.org/datasets/]. The raw CDR data are protected and are not available due to data privacy laws. The processed CDR may be made available by the authors upon reasonable request.

## Code availability

The code for the analysis was programmed using Python 3.6 with standard packages. All the calculations can be reproduced with the equations provided in the main text or the Supplementary Information. The code is available in[41].

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

## Acknowledgements

This work was supported by funding from the US Army Research Office (W911NF-18-1-0421). J.B. acknowledges support by Google Open Source under the Open-Source Complex Ecosystems And Networks (OCEAN) project.

## Author contributions

G.G., R.M., and J.B. developed the concepts and designed the study. Z. C. and S.K. analyzed the data and computed the entropic measures. B.W., and A.E. contributed to the work methodology. A.E. provided the mobile phone data. G.G. and R.M. coordinated the study. All authors contributed to writing the manuscript. Z.C. and S.K. should be considered joint first authors.

## Competing interests

The authors declare no competing interests.
