## [Peer Review File · Nature Communications]

Reviewers' comments:

Reviewer #1 (Remarks to the Author):

The authors use an information-theoretic approach to study how an individual's movements can predict another's. They show that, given a portion of an individual trajectory, information contained in other trajectories, in particular those of this person's social ties, can be helpful to predict his/her future movements.

The topic presented in the article is definitely of major interest, especially because results could have important implications for the anonymization of mobility data. However, I can not recommend publication in a high-impact venue as Nature Communications, as I have two major concerns.

My first concern relates to the data. I am convinced that the limitations of the data, partly presented in the discussion, are severe, to the extent that the interpretation of results is compromised. The results of the paper are largely based on the distinction between social and non-social ties, but the data does not allow to distinguish well between the two. There is no guarantee that pairs of users that are connected in the datasets know each other in person, nor there is guarantee that pairs of users that know each other in person are connected in the dataset.

Secondly, data from LBSN is biased towards specific locations, places where users 'check-in', but it poorly describes overall mobility. While LBSN data was widely used in the early studies on Human Mobility, it is now more common to use other data sources that do not suffer from the same biases (at least for studying "human mobility" in general). In this sense, I would at least expect the authors to better frame their work, by showing the number of locations per day, and the type of points-of-interest covered by the different datasets.

My second concern is methodological. The main message of the paper is that there is some amount of shared information between the trajectories of any given individual "the ego", and those of a subset of other individuals, "the alters". The problem is that, for each ego, the alters are selected in such a way (see section S2) that, by definition, there is some shared information between the two sequences. In this respect, the results are not surprising. Note that, in ref. [20], where the authors use a similar methodology, the analysis is based purely on the ego network of social contacts (there is no match based on similarity of the data). A way to account for how the ego-network is selected could be to compare results with those obtained using the uncorrelated definition of entropy. The way the ego is matched with the alters would be enough to explain the fact that there is some level of overlap between the sets of visited places. But not the order in which places are visited. Further comparison with baseline models which account for spatial constraints would strengthen the paper and make the message more convincing. For example, the authors could study the cross-entropy with realistic synthetic trajectories based around the ego's home-location, where visitation is based on popularity of the locations.

Minor comments:

Fig.1: Trajectories suffering from extremely low data quality (entropy ~ 0) should be filtered out by pre-processing. They can not be used as a proxy of mobility behaviour.

Fig 2:I think x and y labels could be more clear.

Reviewer #2 (Remarks to the Author):

The paper discusses an in-depth analysis of three mobility datasets based on geo-tagged check-ins or location-based social network. The authors apply information-theory measures to the data with the aim of understanding and identifying the sources of predictability in the data, i.e., which data, or combination of data, may lead to good predictions of an individual movement.

The analyses proposed by the authors are interesting and could be potentially used to boost a number of other applications. Their findings suggest new possible ways of encoding additional information in mobility trajectories. The paper is written in good English and the figures are of good quality and readable.

I have several reservations with the overall findings of the paper and with the overall structure of the paper.

In terms of content, my main doubts are:

Fig2 shows several plots regarding cross entropy and cross predictability in one of the three datasets. The authors write that the findings are consistent with that seen for the other two datasets (in the supplementary material). However, it seems to me that cross predictability behaves quite differently, especially for the Gowalla dataset. Some comments regarding this point would be helpful to understand the differences.

The authors in the last part of their discussion, tackle the problem of how time influence cross-predictability. They solve this problem by changing, if I understood correctly, by considering colocations also for individuals with a time-lag. However, it is not clear to me if this lag is considered only for alters also for simple colicators. Moreover, the analysis is performed only on one dataset, and no further material can be found in the supplementary. Also authors indicate a figure, Fig 5B, when, probably, the correct figure for this analysis is Fig 6B.

Still regarding the time problem, I feel that more analyses should be performed, searching for cutoffs to the time-lags to understand if indeed the stability of cross entropy holds and for how long. Some algorithms for the preprocessing of trajectories summarize points up to a window of several hours. So this analysis should be conducted more thoroughly.

In terms of presentation I think that the paper must be improved considerably:

First, the paper is poorly structured: just 4 sections, including introduction, barely any subsection. This makes it hard to pinpoint where to find information for cross referencing and reading the paper. Subdividing the findings of the paper into more sections is advised.

There is no section dedicated to related literature works. Many are indicated in the introduction or as the authors introduce the various measures. Maybe a small section would help in framing the article. The initial part of the paper is disorganized: the authors start by presenting the experimental data, together with mathematical formulation for the data. This should really be separated: first, introduce your mathematical notation, then information about your actual data.

Before each experiment, the authors introduce the metrics, briefly explain them and apply them. Maybe a dedicated section to the mathematical formulation could help the reader in identifying the key meanings of each measure in relation to each other.

Finally, in the conclusions, it would have been nice to have some table or other form of summary of the findings, such as for example, what is the general cut-off for non-social colicators to consider, or other info like that.

Finally It is hard to understand the usefulness of some supplementary material, as they are difficult to read (Fig S3 and S4) or scarcely commented (Table S3)

Overall, I find the paper interesting, but it needs more work.

Reviewer #3 (Remarks to the Author):

In this paper the authors show that one's future location can be predicted by their friends and, more importantly, people they have never even met. It's a strong and interesting message. I appreciate the analysis and the message of this paper, but I have a few major concerns.

Datasets are very old for internet/social media standards. I imagine the way people interact with LBSNs has changed considerably in the last decade. What about geotagged or place-tagged tweets? It would be fascinating to see how these results have changed over time, especially at key epochs (e.g., the Snowden revelations in 2013). However, I think that's a big ask and outside the scope of this work. I would ask that they try to accumulate a more recent dataset.

Figure 1 also calls into question the nature of the BrightKite dataset. Why is the distribution so much different than the other two? A sentence about how most users visit 1-3 places is offered, but I'm not sure what this means. More explanation is needed here.

Requiring 150 check-ins is strict. Indeed it results in a substantial reduction of the data (~90% in some cases). I wonder how these results would hold up on the users with less data. Adding this analysis to the paper would show robustness in results.

The information theoretic results are interesting and useful, but what would really complement the analysis well would be to build (even a simple) classifier to show the accuracy of these different information sources. Ultimately, this is what we really care about, right? Indeed, accuracy is mentioned in the introduction but not in the analysis. Being able to say "hey, we can predict XX% of your future locations" based upon your colocators" is a more direct and powerful message.

Minutia:

"We assume a user who visits N_u locations in total, visits $n_u \leq N_u$ distinct locations." This is not so much an assumption as a tautology.

Reviewer #1 (Remarks to the Author):

The authors use an information-theoretic approach...to study how an individual's movements can predict another's. They show that, given a portion of an individual trajectory, information contained in other trajectories, in particular those of this person's social ties, can be helpful to predict his/her future movements. The topic presented in the article is definitely of major interest, especially because results could have important implications for the anonymization of mobility data.

We are delighted that the referee finds our work of interest and thank them for pointing out the potential and important implications.

My first concern relates to the data. I am convinced that the limitations of the data, partly presented in the discussion, are severe, to the extent that the interpretation of results is compromised. The results of the paper are largely based on the distinction between social and non-social ties, but the data does not allow to distinguish well between the two. There is no guarantee that pairs of users that are connected in the datasets know each other in person, nor there is guarantee that pairs of users that know each other in person are connected in the dataset.

We respectfully disagree. Certainly, inferring social relationships from online data is a challenging enterprise. However, in the datasets considered here, there is a corresponding social network, where users identify each other as acquaintances. In particular, the one's identified as social-ties in our analysis, were reciprocal acquaintances which is a strong marker of an existing social-tie, so a false-positive effect is negligible. There is the issue of false-negatives, where otherwise acquainted users may not identify as such within the platform, but this is a problem specific to every single analysis on such data. Indeed, if one were to restrict analysis to a single data-source, such concerns may well be valid (though practically impossible to quantify). However, as a robustness check, **we analyzed three distinct datasets, and found consistent results across all three.**

Nevertheless, we have now added analysis on a **new mobile phone dataset** (CDR data from Rio de Janeiro, Brazil). There are standard methods for establishing social-ties from such records, but we were particularly strict in how we demarcated between social and non-social ties. For the 30-week period of the data, we marked a social tie as reciprocal calls at least once for all 30 weeks. For non-social ties, we enforced the condition that not a single call was made between two people for the full 30-week period. We reach the same conclusions with these data as with the LBSNs, underscoring the robustness of our study. See more below. We comment upon the methodology on the last para of page 5:

While the LBSN datasets contain explicit social network information, the social network needs to be inferred for the mobile phone data. Following the methodology outlined in [32], we consider *reciprocal* ego-alter pairs as *social-ties*, where reciprocity refers to the fact that both egos and alters exchange phone-calls. We make the reasonable (and stringent) assumption that a reciprocal call should occur at least once a week, and therefore set a minimum threshold of 30 reciprocal calls (covering the period of data-collection) in order to be considered a social-tie. For non-social co-locator networks we require that zero calls exist between the ego and alter.

Secondly, data from LBSN is biased towards specific locations, places where users 'check-in', but it poorly describes overall mobility. While LBSN data was widely used in the early studies on Human Mobility, it is now more common to use other data sources that do not suffer from the same biases (at least for studying "human mobility" in general). In this sense, I would at least expect the authors to better frame their work, by showing the number of locations per day, and the type of points-of-interest covered by the different datasets.

Each data-source on human mobility suffers from its own advantages and limitations (see our work on H. Barbosa et al., Human Mobility: Models and Applications, Phys. Rep. 734, 1-74). LBSNs have the advantage that they are open-source, and therefore the results are reproducible. In terms of resolution, they are better than CDR data, although the coverage may be biased by the user-base. Yet, they show comparable statistical trends to other flavors of mobility datasets in terms of standard metrics such as the distribution of jump-lengths, number of distinct locations visited and the radius of gyration as we clearly show in Fig. S2.

Nevertheless, the point made by the referee in terms of coverage biases is well-taken. For the sake of completeness, as mentioned before, we've included in this new version an analysis of a mobile phone dataset from Rio de Janeiro in Brazil. Rio is a rather large city, and the mobile phone company has wide coverage of the population. This should alleviate any concerns about coverage biases. On the other hand, the problem with CDR data is their lower spatial resolution. Large areas may be covered by a single cell-tower, for instance.

At this point, it is worth emphasizing the following: we now have four datasets, each of which have their own advantages and limitations and inherent biases. But as we show in our revised submission (each figure in the manuscript has been augmented by the CDR analysis), **the main results and trends are the same across all data sources**. All figures have been augmented by the results from the CDR data.

We comment upon the advantages and limitations of each of the datasets on page 19:

Each considered dataset has its own biases related to the behavior of the users, the extent of coverage, and the spatial resolution. The study relied on observational data taken which introduces crucial caveats. In particular, the social ties reported in the datasets are incomplete reflections of a person's full social circle, and the nature of such ties may differ in the online and offline domains. Likewise, not all locations visited by an individual are recorded in these social networks, which rely primarily on check-ins and phone-calls, and as such mobility trajectories may be under-sampled. Followup work, including richer, more detailed data and even experimental studies, are needed to address these concerns, yet our robustness checks, including observing consistent trends across datasets with varying degrees of coverage, resolution, behavioral differences in the observed populations, and across sampling/filtering criteria (see section S7), already provide rather strong evidence for the robustness of our results.

My second concern is methodological. The main message of the paper is that there is some amount of shared information between the trajectories of any given individual "the ego", and those of a subset of other individuals, "the alters". The problem is that, for each ego, the alters are selected in such a way (see section S2) that, by definition, there is some shared information between the two sequences. In this respect, the results are not surprising.

We are assuming that this is a reference to Section S2.1 in the Supplementary Material? First, we note, that the fundamental hypothesis of our paper is that the more frequently two people co-locate, the more likely there is shared information between them. The referee brought up Ref[19] (now Ref[21]). Note in that paper the alters were ranked according to the frequency of engagement with the tweets. This is the same, frequency of engagement = frequency of co-location (the difference being of course in that case it was a single tweet, here they may be multiple locations, or tweets if you will). With this criterion there is **no assumption on the sequence of visits**. Indeed, we contend that the results are indeed rather surprising. Just because two people visit the same grocery store at the same time, does not immediately imply that one person contains information on the other. In fact, that is rather unexpected and is the very point of the paper. In particular, we would like to point the referee's attention to curves in Fig. 2F that include the both the past information of the ego and the alters (effectively the transfer entropy). Fitting a non-linear saturating function to these curves indicate an information gain of ~20% when including alters (for both

social and non-social ties) **above and beyond the information provided by the egos themselves**. We contend that this is a rather surprising result. We include this in the last para of page 12.

Including the ego's past trajectory, for Weeplaces we find $\Pi_\infty = 56.70\%$ and 56.25% for social ties and colocators, respectively. Compared to the average $\Pi_{ego} = 47.05\%$ all egos in the network, this means an *additional* 19.5% to 20.5%, respectively, of the *potential* predictability of an ego is in principle available when including its alters. The closeness of these values underscore the high degree of predictive information available in the non-social colocators. The corresponding findings for the other three datasets are shown in Table S3. In all cases we see a gain of predictability in the ego when alters are included in the range of 2 – 10%.

Granted to get as clean a signal as possible, we filtered out co-locators with zero cross-entropy, however that has a very limited effect on the results. To check for this, we relaxed the filtering process and repeated the experiments for Weeplaces and the mobile phone dataset, **ranking co-locators based on frequency of co-location only. We did no matching with either the correlated or the uncorrelated entropy**. The relevant section on this in the SM (Sec S2.1) has been updated:

We enforce the reasonable constraint that egos and alters (whether social or non-social ties) must co-locate at least more than once across the temporal history of the datasets. Correspondingly we discard all ego-alter pairs that either do not co-locate or co-locate only once and keep the rest. We note that if one were to make a random guess on an ego's next location, at worst that is equivalent to $1/N_{ego}$ where N_{ego} is the number of unique locations in their historical trajectory. For purposes of statistical significance we partition the alter set into those whose cross-entropy is non-zero and those who provide information on an ego equivalent to random guessing, that is $\log_2(N_{ego})$ bits. We term the former "informative" and the latter "non-informative". In the manuscript we present results derived from the "informative" alters, in Sec. S7 we show the results when also including the "non-informative" alters.

In a new section in the SM, titled Robustness Analysis (Section S7). Dropping the filtering constraint leads to larger networks of size 827 (Weeplaces) and 1103 (CDR) egos who have at least 10 co-locators in both types of networks. In Fig. S23 we plot the equivalent of Fig. 2F (cross-predictability as a function of accumulated alters) for Weeplaces A and the mobile phone dataset B. For the case of Weeplaces, while we see about a 10% drop in $\Pi_{A|B}$ for both the social and non-social ties as compared to Figure 2F, the qualitative trends remain robust. That is, as alters are accumulated we see a corresponding gain on information in the ego. We find the same result for the mobile phone dataset (panel B) with now a more modest drop of 5%. Nevertheless, once again the monotonically increasing trend of the cross-predictability via alter accumulation remains robust. **Thus despite not imposing any matching on alter-ego pairs based on any entropy measure, we see the same trends.**

In the same section, we also relax the number of check-ins from 150 to 75, as well as alter the temporal window. In all cases, the results continue to remain robust. Further, as we show in Fig 6 and S19, we look at the case for time-displaced co-locators, spanning from 0.5H-12H. Panel B of Fig. S20 shows that the networks in each case have very little overlap, yet we continue to find the same trends! We believe that taken together--the new dataset, no matching of alter-ego pairs, robustness to temporal windows and offsets---makes an exceptionally strong case for the strength of our results.

A way to account for how the ego-network is selected could be to compare results with those obtained using the uncorrelated definition of entropy. The way the ego is matched with the alters would be enough to explain the fact that there is some level of overlap between the sets of visited places. But not the order in which places are visited.

As discussed above, we repeated the experiment with no matching at all, and the results remain robust.

Further comparison with baseline models which account for spatial constraints would strengthen the paper and make the message more convincing. For example, the authors could study the cross-entropy with realistic synthetic trajectories based around the ego's home-location, where visitation is based on popularity of the locations.

This is a great suggestion. And augmenting our data with modeling fits in well with another reviewer comment about applying machine learning prediction to augment our information-theoretic bounds. However, it brings a lot of additional complexity—selecting, calibrating, validating, and comparing mobility models— so we feel that this deserves its own paper. We have presented a number of results in this manuscript, as such it is already quite dense.

Fig.1: Trajectories suffering from extremely low data quality (entropy ~ 0) should be filtered out by pre-processing. They cannot be used as a proxy of mobility behaviour.

Indeed, we are in complete agreement that the low entropy users are a poor proxy for mobility. However Figure 1 shows the raw data prior to construction of the ego-alter networks. It serves to show the diversity of behavior between the datasets. None of the low-entropy users are present in the subsequent analysis post Figure 1.

Fig 2: I think x and y labels could be more clear.

We will be happy to do so, however we are uncertain as to whether this is a reference to the size of the labels, or is it that we should use text-labels along with the mathematical symbols. Will change pending clarification.

All changes made to the manuscript are marked in red for convenient lookup. Do note the addition of a co-author, Alexandre Evsukoff, who helped with obtaining the CDR data and the subsequent analysis.

In summary, we thank the referee for the constructive comments, and in particular prompting us to consider other types of datasets. The suggestions have improved the manuscript immensely. We look forward to a favorable assessment.

Reviewer #2 (Remarks to the Author):

The analyses proposed by the authors are interesting and could be potentially used to boost a number of other applications. Their findings suggest new possible ways of encoding additional information in mobility trajectories. The paper is written in good english and the figures are of good quality and readable.

We thank the referee for the positive assessment.

Fig2 shows several plots regarding cross entropy and cross predictability in one of the three dataset. The authors write that the findings are consistent with that seen for the other two datasets (in the supplementary material). However, it seems to me that cross predictability behaves quite differently, especially for the Gowalla dataset. Some comments regarding this point would be helpful to understand the differences.

We apologize for this oversight. We are remiss in not discussing each of the cases more thoroughly but do note we are constrained by the journal constraints on just how much can be presented in the main manuscript. Nevertheless, where applicable we have elaborated more on the results from the other datasets. For instance on last para of page 4:

In Fig. S1 we show the check-in maps for each of the datasets indicating global coverage with the highest concentrations in North America and Western Europe. In Fig. S2 we plot the distributions for the number of distinct locations visited by users, the jump-length and the radius-of-gyration. The LBSNs show similar trends for their jump-length and radius-of-gyration, consistent with other sources of mobility data [1]. Differences exist, however, in the number of unique locations visited. BrightKite, in particular, contains a large fraction of users that visit only a few distinct locations (between 1 and 3). The CDR data differs from the LBSN in having much sharper cut-offs in the tail of the distribution, a consequence of scale—trips are bounded by the spatial area of the city, unlike in the LBSNs that contain inter-city and international travel. The statistical trends, however, are consistent with that seen in sources of mobile phone data [31].

First para page 12:

The corresponding results for the other three datasets are shown in Fig. S6 (BrightKite), Fig. S7 (Gowalla) and Fig. S8 (CDR). While the values of the entropies, predictability, cross-predictability and the comparable information contained in the top social tie viz the number of non-social colocators vary between the datasets (reflecting their inherent biases), in all cases the trends are consistent with that seen in Fig. 2.

The differences and biases between the datasets and their effects on the results has been added throughout the manuscript where appropriate (marked in red).

The authors in the last part of their discussion, tackle the problem of how time influence cross-predictability. They solve this problem by changing, if I understood correctly, by considering colocators also for individuals with a time-lag. However, it is not clear to me if this lag is considered only for alters also for simple colocators. Moreover the, the analysis is performed only on one dataset, and no further material can be found in the supplementary.

The time-lag does not refer to egos or alters, but is used to define simple co-locators. Once this is established, then the corresponding ego-alter network is constructed as before.

Also authors indicate a figure, Fig 5B, when, probably, the correct figure for this analysis is Fig 6B.

This is correct, thank you for pointing this out, we corrected the reference to the figure.

Still regarding the time problem, I feel that more analyses should be performed, searching for cutoffs to the time-lags to understand if indeed the stability of cross entropy holds and for how long. Some algorithms for the preprocessing of trajectories summarize points up to a window of several hours. So this analysis should be conducted more thoroughly.

We were remiss in not doing so. We have now conducted the analysis on a new dataset based on mobile phone records (included to address the concerns of some of the other referees). We are unable to do so for Brightkite and Gowalla, since they do not contain enough common egos who have at least 10 alters in all 24 one Hr-lag networks. In Fig. S20 and S21 we now extend the analysis for both datasets in the range of time-lags of 0.5H-12H finding the same trends as reported in Fig. 6 in the main manuscript.

*In terms of presentation I think that the paper must be improved considerably:
First, the paper is poorly structured: just 4 sections, including introduction, barely any subsection. This makes hard to pinpoint where to find information for cross referencing and reading the paper.
Subdividing the findings of the paper into more sections is advised.*

We agree with the referee, the use of subsections will make things much easier to understand, and we have now added those.

*There is no section dedicated to related literature works. Many are indicated in the introduction or as the authors introduce the various measures. Maybe a small section would help in framing the article.
The initial part of the paper is disorganized: the authors start by presenting the experimental data, together with mathematical formulation for the data. This should really be separated: first, introduce your mathematical notation, then information about your actual data.
Before each experiment, the authors introduce the metrics, briefly explain them and apply them. Maybe a dedicated section to the mathematical formulation could help the reader in identifying the key meanings of each measure in relation to each other.
Finally, in the conclusions, it would have been nice to have some table or other form of summary of the findings, such as for example, what is the general cut-off for non-social colocators to consider, or other info like that.*

All of these concerns raised by the referee are understandable, and the suggestions are reasonable, Thank you. Our goal was to write in conformance with the style guidelines of the journal. If at the editorial stage, there is a request to alter the format, then we will be very happy to do so.

Finally It is hard to understand the usefulness of some supplementary material, as they are difficult to read (Fig S3 and S4) or scarcely commented (Table S3)

We have updated the figures for clarity. The results from Table S3 are extensively commented upon on page 12 in the subsection *Combined information of both ego and alter's past*.

This saturation effect is examined in Sec. S4, where $\Pi_A|B$ is extrapolated beyond our data window of 10 alters by fitting a nonlinear saturating function and estimating the extrapolation predictability Π_∞ . For Weeplaces we find $\Pi_\infty = 44.32\%$ and 39.79% for social ties and colocators, respectively. Compared to the average $\Pi_{ego} = 47.05\%$ all egos in the network, this means that 94% and 85%, respectively, of the *potential* predictability of an ego is in principle available in that ego's alters. Including the ego's past trajectory, for Weeplaces we find $\Pi_\infty = 56.70\%$ and 56.25% for social ties and colocators, respectively. Compared to the average $\Pi_{ego} = 47.05\%$ all egos in the network, this means an *additional* 19.5% to 20.5%, respectively, of the *potential* predictability of an ego is in principle available when including its alters.

All changes in the manuscript are marked in red for easy look-up. Do note the addition of a co-author, Alexandre Evsukoff, who helped with obtaining the CDR data and the subsequent analysis.

In summary we thank the referee for the constructive comments and prompting us to clarify aspects of the manuscript. We hope for a favorable assessment.

Reviewer #3 (Remarks to the Author):

In this paper the authors show that one's future location can be predicted by their friends and, more importantly, people they have never even met. It's a strong and interesting message. I appreciate the analysis and the message of this paper.

We thank the referee for finding the paper of interest, and appreciating the analysis.

Datasets are very old for internet/social media standards. I imagine the way people interact with LBSNs has changed considerably in the last decade. What about geotagged or place-tagged tweets? It would be fascinating to see how these results have changed over time, especially at key epochs (e.g., the Snowden revelations in 2013). However, I think that's a big ask and outside the scope of this work. I would ask that they try to accumulate a more recent dataset.

This is a fair point, however each type of mobility data comes with its own advantages and disadvantages. Geo-located tweets are noisy and incomplete, GPS data while being the most precise are hard to obtain and not scalable, Mobile phone (CDR) data provides good coverage but has poor spatial resolution and is moreover rarely open-access. LBSN data splits the differences between these types, having decent spatial resolution, is scalable and most importantly has many examples of open-access sources. Moreover, the LBSN datasets also contain information on the social network, which is crucial for our analysis. Of course the mobility patterns might be biased by the type of users using the platform. Note, however we use three different datasets and find consistent results which attests to the robustness of the analysis.

Nevertheless, we take the referee's criticism in the right spirit, and rather than collect another example of LBSN data, we conduct our analysis on an entirely different type of dataset. We analyzed a **Mobile Phone CDR data from Rio de Janeiro in Brazil**, that has comprehensive coverage of the resident population. We constructed the social network of phone users using established methods based on the calls made between individuals. All figures have been augmented with the CDR analysis, and we obtain the same results.

Figure 1 also calls into question the nature of the BrightKite dataset. Why is the distribution so much different than the other two? A sentence about how most users visit 1-3 places is offered, but I'm not sure what this means. More explanation is needed here.

As we show in Fig. S1, BrightKite is unusual among the datasets in that it has a large fraction of its users visit between 1-3 distinct locations. If somebody only visits one location then their entropy is 0 and their predictability 1. So users that visit such few locations will in general have very low values of entropy and correspondingly high values of predictability. Since such a large fraction of them exist in BK, we see the peaks near $S=0$ and $\text{PI} = 1$ for BK. Do note that this does not bias any of the subsequent results, as such low-entropy users do not exist in the ego-alter social networks.

Requiring 150 check-ins is strict. Indeed it results in a substantial reduction of the data (~90% in some cases). I wonder how these results would hold up on the users with less data. Adding this analysis to the paper would show robustness in results.

We thank the referee for bringing up this very important point. This is indeed something we should have done unprompted. We now include an analysis for Weeplaces and the CDR data for 75 check-ins, a much less stringent condition. The results are plotted in a new Section S7, titled Robustness analysis. As can be seen from Fig. S21, there is very little difference between the two criteria.

We have also added analysis for much larger networks consisting of ~1000 egos for Weeplaces and the mobile phone data in Fig. S23, showing similar results.

The information theoretic results are interesting and useful, but what would really complement the analysis well would be to build (even a simple) classifier to show the accuracy of these different information sources. Ultimately, this is what we really care about, right? Indeed, accuracy is mentioned in the introduction but not in the analysis. Being able to say "hey, we can predict XX% of your future locations" based upon your colocators" is a more direct and powerful message.

This is an excellent point because it moves our theoretical analysis out to the real world. However, the world of machine learning is moving incredibly quickly, and what is state-of-the-art today may be obsolete tomorrow. So, at best a result along these lines can provide only a lower-bound on what can be achieved. And in the case of a simple classifier, it will likely be a very loose bound! Therefore, we tackle the problem from above, asking the mathematical ideal degree to which prediction can be made. While less practical in the short-term, this viewpoint has the advantage of proving a bound on *all possible methods* applied to these data and *cannot become obsolete*. In our opinion, this is also a powerful message.

Minutia:

"We assume a user who visits N_u locations in total, visits $nu \leq N_u$ distinct locations." This is not so much an assumption as a tautology.

We have revised the sentence to eliminate the tautology and focus on our goal of defining notation. Thank you.

All changes in the manuscript are marked in red for easy look-up. Do note the addition of a co-author, Alexandre Evsukoff, who helped with obtaining the CDR data and the subsequent analysis.

In summary, we thank the referee for the constructive comments, in particular prompting us to conduct the analysis on a new dataset, as well as the robustness analysis on applying a less stringent filtering condition. We look forward to a favorable assessment.

REVIEWER COMMENTS

Reviewer #1 (Remarks to the Author):

First of all, I would like to stress again that I find that the topic addressed in this article is compelling. From a theoretical standpoint, it investigates to what extent mobility traces are similar across individuals. This understanding has important practical consequences for addressing data anonymization and for modelling purposes. Further, under many aspects this is a manuscript of high quality standards. Analyses are extensive and well presented. The inclusion of the CDR dataset makes the study more convincing.

At the same time, I still have concerns regarding the methodology, and I do not understand how the analyses included in the SI section addresses my doubts. I will now try to reformulate my concerns more explicitly. This is a question that other readers may have, so I hope that the authors can explain more clearly how the new robustness analyses address these doubts.

My question is the following: Could the observed results simply be due to the fact that the "ego-alter" pairs have at least one location in common?

Let me explain this more clearly.

The authors choose the alters in such a way that there is at least one common location between the ego and the alter. Now, given the tendency of people to return to previously visited locations, it is likely that this "common" location will be visited more times by both the ego and the alter.

Given that the ego and the alter trajectories share at least one location, I expect that in general $S(\text{ego}|\text{ego+alter}) < S(\text{ego}|\text{ego})$, because the alter may visit the common location for the first time before the ego.

The more locations in common, the more this effect should be pronounced.

The more alters are included, the more this effect should be pronounced.

Let me explain this better with an example.

Let's say the ego trajectory is a sequence of N visits to L distinct locations. Let's say at each time, one location from the set of distinct locations is chosen with uniform probability.

Let's say the alter trajectory is also a sequence of N visits to L distinct locations, picked with uniform probability. But let's assume that there are O common locations between the ego and the alter.

By simulating this example for 10000 random sequences, with $N=300$, $L=5$, I find the following results for $S(\text{ego}|\text{ego})$ vs $S(\text{ego}|\text{alter})$ for increasing values of O .

overlap = 0

$S(\text{ego}|\text{ego}) = 1.5343 \pm 0.0001$

$S(\text{ego}|\text{alter,ego}) = 1.5343 \pm 0.0001$

overlap = 1

$S(\text{ego}|\text{ego}) = 1.5345 \pm 0.0001$

$S(\text{ego}|\text{alter,ego}) = 1.5324 \pm 0.0001$

overlap = 2
 $S(\text{ego}|\text{ego}) = 1.5346 \pm 0.0001$
 $S(\text{ego}|\text{alter,ego}) = 1.5244 \pm 0.0001$

overlap = 3
 $S(\text{ego}|\text{ego}) = 1.5342 \pm 0.0001$
 $S(\text{ego}|\text{alter,ego}) = 1.5025 \pm 0.0001$

overlap = 4
 $S(\text{ego}|\text{ego}) = 1.5343 \pm 0.0001$
 $S(\text{ego}|\text{alter,ego}) = 1.4586 \pm 0.0001$

Now, let's consider the case where there are multiple alters. I picked 10000 traces, $L=5$, $N=300$, and $O=3$ (but the overlap location is picked at random). I have the following result for increasing numbers of alters.

alters = 0
 $S(\text{ego}|\text{ego}) = 1.5342 \pm 0.0001$
 $S(\text{ego}|\text{alters,ego}) = 1.5342 \pm 0.0001$

alters = 1
 $S(\text{ego}|\text{ego}) = 1.5342 \pm 0.0001$
 $S(\text{ego}|\text{alters,ego}) = 1.5028 \pm 0.0001$

alters = 2
 $S(\text{ego}|\text{ego}) = 1.5343 \pm 0.0001$
 $S(\text{ego}|\text{alters,ego}) = 1.4774 \pm 0.0001$

alters = 3
 $S(\text{ego}|\text{ego}) = 1.5348 \pm 0.0001$
 $S(\text{ego}|\text{alters,ego}) = 1.4568 \pm 0.0001$

alters = 4
 $S(\text{ego}|\text{ego}) = 1.5344 \pm 0.0001$
 $S(\text{ego}|\text{alters,ego}) = 1.4391 \pm 0.0001$

The trend above is similar to the one presented in the paper, even if the sequences are random. Of course, these are simple trajectories with a few locations. But I hope this makes it clear why I ask myself the question: Could the observed results simply be due to the fact that the "ego-alter" pairs have at least one location in common? When people have one or more locations in common, the two traces share more information irrespectively of the order of locations (in the example above the order is random).

Other points:

The authors could check if there is a difference in shared information across pair of individuals that co-locate vs pairs of individuals that simply have locations in common. The authors could better clarify the distinction between informative and non-informative individuals (supplementary section S7). How do you find individuals that provide information equivalent to random guessing? What is the reason why they are excluded from

the analysis in the main text?

A minor point: When you say individuals with non-zero cross-entropy, do you mean individuals that have more than one visit? Or to say it better, who are the pairs of individuals with zero cross-entropy?

Final point:

To ensure reproducibility, the authors should share their code.

The code I used is attached to the letter (and pasted below). I hope I interpreted well the method described in the paper.

First of all, I would like to stress again that I find that the topic addressed in this article is compelling. From a theoretical standpoint, it investigates to what extent mobility traces are similar across individuals. This understanding has important practical consequences for addressing data anonymization and for modelling purposes.

Further, under many aspects this is a manuscript of high quality standards. Analyses are extensive and well presented. The inclusion of the CDR dataset makes the study more convincing.

At the same time, I still have concerns regarding the methodology, and I do not understand how the analyses included in the SI section addresses my doubts. I will now try to re-formulate my concerns more explicitly. I guess this is a question that other readers may have, so I hope that the authors can explain more clearly how the new robustness analyses address these doubts.

My question is the following: **Could the observed results simply be due to the fact that the “ego-alter” pairs have at least one location in common?**

Let me explain this more clearly.

- 1) The authors choose the alters in such a way that there is at least one common location between the ego and the alter. Now, given the tendency of people to return to previously visited locations, it is likely that this “common” location will be visited more times by both the ego and the alter.
- 2) Given that the ego and the alter trajectories share at least one location, I expect that in general $S(\text{ego}|\text{ego+alter}) < S(\text{ego}|\text{ego})$, because the alter may visit the common location for the first time before the ego.
- 3) The more locations in common, the more this effect should be pronounced.
- 4) The more alters are included, the more this effect should be pronounced.

Let me explain this better with an example.

Let's say the ego trajectory is a sequence of N visits to L distinct locations. Let's say at each time, one location from the set of distinct locations is chosen with uniform probability.

Let's say the alter trajectory is also a sequence of N visits to L distinct locations, picked with uniform probability. But let's assume that there are O common locations between the ego and the alter.

By simulating this example for 10000 random sequences, with $N=300$, $L=5$, I find the following results for $S(\text{ego}|\text{ego})$ vs $S(\text{ego}|\text{alter})$ for increasing values of O .

Now, let's consider the case where there are multiple alters. I picked 10000 traces, $L=5$, $N=300$, and $O=3$ (but the overlap location is picked at random). I have the following result for increasing numbers of alters.

The results above are very similar to the ones presented in the paper, even if the sequences are random. Of course, these are simple trajectories with a few locations. But I hope this makes it clear why I ask myself the question: **Could the observed results simply be due to the fact that the “ego-alter” pairs have at least one location in common?** When people have one or more locations in common, the two traces share more information irrespectively of the order of locations (in the example above the order is random).

Other points:

- 1) The authors could check if there is a difference in shared information across pair of individuals that co-locate vs pairs of individuals that simply have locations in common.
- 2) The authors could better clarify the distinction between informative and non-informative individuals (supplementary section S7). How do you find individuals that provide information equivalent to random guessing? What is the reason why they are excluded from the analysis in the main text?
- 3) A minor point: When you say individuals with non-zero cross-entropy, do you mean individuals that have more than one visit? Or to say it better, who are the pairs of individuals with zero cross-entropy?

Final point:

To ensure reproducibility, the authors could share their code.

The code I used is attached to the letter (and pasted below). I hope I interpreted well the method described in the paper.

```
from collections import defaultdict
import numpy as np
import random
from joblib import Parallel, delayed
import math
import matplotlib.pyplot as plt

def contains(small, big):
    try:
        big.tobytes().index(small.tobytes())//big.itemsize
        return True
    except ValueError:
        return False

def cross_entropy(l1,l2):
    #l1 is an array
    #l2 is array of arrays (can include many alters)
    n = len(l1)
    n2 = np.mean([len(i) for i in l2])

    sum_gamma = 0

    for i in range(0, n):
        sequences = [l[:i] for l in l2]

        for j in range(i+1, n+1):
            s = l1[i:j]
            if any(contains(s, sequence) for sequence in sequences) != True:
                sum_gamma += len(s)
                break

    ae = 1 / (sum_gamma / n) * math.log(n2)
    return ae
```

```

def cross_entropy_for_two_random_trajectories(n_locations,
                                             sequence_length,
                                             overlap_length):

    #N_locations is the number of unique locations
    #sequence_length is the length of the sequences
    #overlap_length is the number of locations in common between ego and alter
    #wait_time_before_overlap is the number of steps to wait before the ego starts visiting the location in common

    ##LOCATIONS AND PROBABILITY OF VISITS
    locations_ego = range(n_locations)
    locations_alter = range(n_locations-overlap_length,n_locations*2-overlap_length)
    probability = [1/n_locations for _ in locations_ego]

    ##BUILD TRAJECTORIES
    sequence_ego = np.random.choice(locations_ego,
                                    p=probability,
                                    size=sequence_length)
    sequence_alter = np.random.choice(locations_alter, p=probability, size=sequence_length)

    ##COMPUTE CROSS ENTROPY
    S_ego = cross_entropy(sequence_ego,[sequence_ego])
    S_ego_alter = cross_entropy(sequence_ego,[sequence_ego, sequence_alter])

    return S_ego, S_ego_alter

```

```

def cross_entropy_for_N_random_trajectories(n_locations,
                                           n_alter,
                                           sequence_length,
                                           overlap_length):

    #N_locations is the number of unique locations
    #sequence_length is the length of the sequences
    #overlap_length is the number of locations in common between ego and alter
    #wait_time_before_overlap is the number of steps to wait before the ego starts visiting the location in common

    ##LOCATIONS AND PROBABILITY OF VISITS
    locations_ego = range(n_locations)
    probability = [1/n_locations for _ in locations_ego]

    ##BUILD TRAJECTORIES
    sequence_ego = np.random.choice(locations_ego,
                                    p=probability,
                                    size=sequence_length)

    ##COMPUTE CROSS ENTROPY
    sequences_alter = []
    for alter in range(n_alter):
        common_locations = np.random.choice(locations_ego,overlap_length,replace=False)
        locations_alter = list(common_locations) + list(range(n_locations,n_locations*2-overlap_length))
        sequence_alter = np.random.choice(locations_alter, p=probability, size=sequence_length)
        sequences_alter.append(sequence_alter)

    S_ego = cross_entropy(sequence_ego,[sequence_ego])
    S_ego_alter = cross_entropy(sequence_ego,[sequence_ego]+sequences_alter)

    return S_ego, S_ego_alter

```

Individuals with and without locations overlap

```

In [91]:
examples = 10000
n_locations = 5
length = 300
n_jobs=5
results = []

for overlap in range(0,n_locations):
    result = Parallel(n_jobs=n_jobs)(delayed(cross_entropy_for_two_random_trajectories)(n_locations,length,overlap) for i in range(examples))
    results.append(result)

```

More and more individuals

In [102]

```
examples = 1000
n_locations = 5
length = 300
n_jobs=5
results = []

for n_alters in range(0,5):
    result = Parallel(n_jobs=n_jobs)(delayed(cross_entropy_for_N_random_trajectories)(n_locations,n_alters,length,overlap) for i in range(examples))
    results.append(result)
```

Reviewer #2 (Remarks to the Author):

The revised manuscript is improved with respect the previous version. The authors provided answers for any raised question and doubt. Some questions were addressed adding additional material and discussion on the paper while other questions were addressed only in the response letter explaining the reason.

I think that the current version of the paper is enough suitable for publication.

Reviewer #3 (Remarks to the Author):

The changes show a substantial improvement in the paper. There are two dangling issues:

- I do not agree that because ML technology is advancing that it is not worthwhile to show us a weak lower bound. Again, this plays to the use of the term "accuracy" throughout.**
- Kudos for adding the CDR data. It's a shame that 2014 is the newest data we can get. Maybe LBSNs are a thing of the past.**

Report of the First Referee

First of all, I would like to stress again that I find that the topic addressed in this article is compelling. From a theoretical standpoint, it investigates to what extent mobility traces are similar across individuals. This understanding has important practical consequences for addressing data anonymization and for modelling purposes. Further, under many aspects this is a manuscript of high quality standards. Analyses are extensive and well presented. The inclusion of the CDR dataset makes the study more convincing.

We thank the referee for the kind words and positive assessment.

At the same time, I still have concerns regarding the methodology, and I do not understand how the analyses included in the SI section addresses my doubts.

In the previous report concerns were raised about the filtering process (informative vs non-informative alters). We demonstrated in the additional analysis presented in the SI, that the unfiltered data showed the same trends.

I will now try to re-formulate my concerns more explicitly. This is a question that other readers may have, so I hope that the authors can explain more clearly how the new robustness analyses address these doubts. My question is the following: Could the observed results simply be due to the fact that the “ego-alter” pairs have at least one location in common?

We thank the referee for taking the time to go through the details of the analysis and providing the code for their own analysis. Indeed, the stated question is much clearer now. The key thing to note here, is that the phenomenon we are investigating is not merely shared information, but predictability. That is, what is the likelihood of guessing an individual’s next location given their prior history? What becomes important then is not just the number of shared unique locations (which is a first order effect), but the specific sequence of visitations. The referee is correct in the observation that increasing the number of shared locations between two individuals decreases their cross-entropy, but this does not necessarily lead to increased predictability.

A simple analysis can illustrate this point. We can take a particular trajectory and then duplicate it. Imagine that this duplicated trajectory represents a perfect alter. The ego and alter share the same number of visited locations and the same sequence. This will lead to a certain cross-predictability. One can then randomize the sequence of the alter multiple times and then compute the average cross-predictability over those trials. Further, one can progressively lower the location overlap, by replacing locations in the duplicated trajectory with alternative ones. In

each instance the same randomization can be done, and a corresponding average cross-predictability computed.

We did this analysis for the top 20% most predictable egos in Weeplaces and the mobile phone dataset (we used the empirical data instead of synthetic trajectories to preserve the distributions of location visits). The figure is appended below:

The **orange horizontal line** represents the baseline cross-predictability preserving the sequence and 100% location overlap (solid line, average, shaded area, 95% CI). The **blue curve** represents the corresponding quantity over 100 randomizations of each sequence. As is abundantly clear from the figure, location overlap on its own is not nearly enough to explain the reported results. Even with a 100% location overlap, randomizing the sequence leads to information loss.

Indeed, one can systematically check for this from the actual ego-alter trajectories in the dataset. For each considered ego, we take its top ranked alter and then randomize its trajectory and re-estimate the predictability. This will generate a distribution of cross-predictabilities with a given mean. One can then formulate a hypothesis of whether the true cross-predictability---derived from the actual alter trajectory--is greater than the average cross-predictability computed by randomizing the alter trajectories. Doing this for the top ten alters of an ego and conducting a one-sided t-test at multiple levels of significance (5% and 1%) leads to the following results for the 4 datasets.

The orange bars refer to the non-social colocators and the blue to the social network. As the figure suggests, across all datasets and in both types of network, for at least 50% of the egos, the information provided by the true sequence significantly exceeds that of the randomized trajectories. In the case of BrightKite and in particular the mobile phone dataset, this is true for between 75-90% of all considered egos. Thus, very clearly the reported effects are not explained just by shared locations.

We include this analysis in Section S7.4 of the SI and refer to it in the main manuscript on page 5, second paragraph.

The authors could better clarify the distinction between informative and non-informative individuals (supplementary section S7). How do you find individuals that provide information equivalent to random guessing?

We agree, we could indeed do a better job, and we rewrote the section, including a specific example of a “non-informative alter”. In Section S2.1:

We enforce the reasonable constraint that egos and alters (whether social or non-social ties) must co-locate at least more than once across the temporal history of the datasets. Correspondingly we discard all ego-alter pairs that either do not co-locate or co-locate only once and keep the rest. We note that if one were to make a random guess on an ego's next location, at worst that is

equivalent to $1/n_{ego}$ where n_{ego} is the number of unique locations in their historical trajectory. The corresponding information provided due to this random guessing is $\log_2(n_{ego})$ bits.

As an example of such users in our database, consider the following: an ego A , and their alter B in the Weeplaces dataset. The number of unique locations visited by A is $n_A = 254$ and the total check-ins are $N_A = 524$. For B the corresponding numbers are $n_B = 250$ and $N_B = 544$. The check-in time period for A is [2009-03-15 01:16:25, 2010-10-18 18:38:33] while that for B is [2009-10-29 02:55:32, 2010-10-21 23:54:30]. Correspondingly $1/n_A = 0.003937$ and $\log_2(254) = 7.9887$.

Although B checked in 541 times while A was active, A only checked in 384 times during the period that B was active. That is to say, the whole check-in sequence of B contributes nothing to the first $524 - 384 = 140$ check-ins of A .

Mathematically speaking, 140 zeros at the beginning of the sum in the denominator of Eq.~\eqref{eq:CCE} result in a relatively large $\hat{S}_{A|B}$. In this example, $\hat{S}_{A|B} = 8.4280 > 7.9887 = \log_2(254)$, therefore alter B 's trajectory provides no more information on A than simple random guessing based on A 's location history.

When you say individuals with non-zero cross-entropy, do you mean individuals that have more than one visit? Or to say it better, who are the pairs of individuals with zero cross-entropy?

We thank the referee for pointing this out. This was an incorrect statement. We meant to write individuals whose cross-entropy is less than $\ln_2(n_{ego})$. This has now been corrected.

To ensure reproducibility, the authors should share their code.

We agree, and we intended to do so. We now include a code availability statement with a link to the code.

In summary we thank the referee for the thoughtful comments and prompting us to clarify our methodology. We feel that the manuscript is much improved for it.

REVIEWERS' COMMENTS

Reviewer #1 (Remarks to the Author):

My comments were addressed.

The manuscript has substantially improved in the rounds of revision and I judge it to be ready for publication.

Response to Reviewer

My comments were addressed.

The manuscript has substantially improved in the rounds of revision and I judge it to be ready for publication.

We thank the referee for endorsing the manuscript.